# Clinicians’ Perspectives on Barriers and Facilitators for the Adoption of Non-Invasive Liver Tests for NAFLD: A Mixed-Method Study

**DOI:** 10.3390/jcm11102707

**Published:** 2022-05-11

**Authors:** Yasaman Vali, Roel Eijk, Timothy Hicks, William S. Jones, Jana Suklan, Adriaan G. Holleboom, Vlad Ratziu, Miranda W. Langendam, Quentin M. Anstee, Patrick M. M. Bossuyt

**Affiliations:** 1Department of Epidemiology and Data Science, Amsterdam Public Health, Amsterdam UMC Location University of Amsterdam, 1105 AZ Amsterdam, The Netherlands; m.w.langendam@amsterdamumc.nl (M.W.L.); p.m.bossuyt@amsterdamumc.nl (P.M.M.B.); 2Athena Institute, Faculty of Science, VU University Amsterdam, 1081 HV Amsterdam, The Netherlands; r.eijk@student.vu.nl; 3NIHR Newcastle In Vitro Diagnostics Co-Operative, Translational and Clinical Research Institute, Newcastle University, Newcastle upon Tyne NE1 7RU, UK; timothy.hicks@nhs.net (T.H.); will.jones@newcastle.ac.uk (W.S.J.); jana.suklan@newcastle.ac.uk (J.S.); 4NIHR Newcastle In Vitro Diagnostics Co-Operative, Newcastle upon Tyne Hospitals Foundation Trust, Newcastle upon Tyne NE1 7RU, UK; 5Department of Internal and Vascular Medicine, Amsterdam UMC Location University of Amsterdam, 1105 AZ Amsterdam, The Netherlands; a.g.holleboom@amsterdamumc.nl; 6Assistance Publique-Hôpitaux de Paris, Hôpital Beaujon, University Paris-Diderot, 75013 Paris, France; vlad.ratziu@inserm.fr; 7The Newcastle Liver Research Group, Translational & Clinical Research Institute, Faculty of Medical Sciences, Newcastle University, Newcastle upon Tyne NE1 7RU, UK; quentin.anstee@newcastle.ac.uk; 8Newcastle NIHR Biomedical Research Centre, Newcastle upon Tyne Hospitals NHS Foundation Trust, Newcastle upon Tyne NE1 7RU, UK

**Keywords:** non-alcoholic fatty liver disease, non-invasive tests, adoption, implementation research, NASSS framework, mixed method

## Abstract

(1) Background: Given the high prevalence of non-alcoholic fatty liver disease (NAFLD) and the limitations of liver biopsies, multiple non-invasive tests (NITs) have been developed to identify non-alcoholic fatty liver disease (NAFLD) patients at-risk of progression. The availability of these new NITs varies from country to country, and little is known about their implementation and adoption in routine clinical practice. This study aims to explore barriers and facilitators that influence the adoption of NAFLD NITs, from healthcare professionals’ perspectives. (2) Methods: A cross-sectional study was performed using an exploratory mixed-methods approach. Twenty-seven clinicians from eight different countries with different specialties filled in our questionnaire. Of those, 16 participated in semi-structured interviews. Qualitative and quantitative data were collected and summarized using the recently published Non-adoption, Abandonment, Scale-up, Spread, and Sustainability (NASSS) framework for new medical technologies in healthcare organizations. (3) Results: Several factors were reported as influencing the uptake of NITs for NAFLD in clinical practice. Among those: insufficient awareness of tests; lack of practical guidelines and evidence for the performance of tests in appropriate patient populations and care settings; and absence of sufficient reimbursement systems were reported as the most important barriers. Other factors, most notably ‘local champions’, proper functional payment systems, and sufficient resources in academic hospitals, were indicated as important facilitating factors. (4) Conclusions: Clinicians see the adoption of NITs for NAFLD as a complex process that is modulated by several factors, such as robust evidence, practical guidelines, a proper payment system, and local champions. Future research could explore perspectives from other stakeholders on the adoption of NITs.

## 1. Introduction

Non-alcoholic fatty liver disease (NAFLD) is an increasingly prevalent, complex, and progressive liver condition, which is presenting a growing challenge to healthcare systems internationally [1,2,3,4]. NAFLD patients are at risk of progression to more severe stages, such as non-alcoholic steatohepatitis (NASH) and/or advanced liver fibrosis and cirrhosis. The progressive nature of this disease and its high global prevalence highlight the importance of timely identification of patients at risk of progression and of assessing the severity of the disease.

Currently, liver biopsy is the reference standard for definitive diagnosis of NAFLD, detecting NASH, and accurately staging liver fibrosis in NAFLD patients. However, it is invasive, has risks of complications, and is subject to sampling variability and inter-observer variation in interpretation [5,6]. These limitations have fuelled interest in developing non-invasive tests (NITs) to evaluate NAFLD progression [3,7,8,9,10]. Multiple biomarker-based NITs have been developed in recent years. Some, like the Enhanced Liver Fibrosis (ELF) test and FibroScan, are advocated by clinical guidelines for assessing fibrosis in NAFLD patients [11]. However, recent surveys detail that the availability of these tests in clinics varies from country to country, influenced by factors such as different regulatory requirements, national policies, and insurance coverage [12,13].

The global need for and interest in new and accurate medical tests is not limited to the hepatology field. The discovery and development of biomarkers have been a highly exciting field of research, as novel markers may have substantial potential to improve health outcomes for a range of medical conditions. Large investments, by both academia and industry, have been made in this area, but the path from the discovery of biomarkers to implementation in innovative medical tests is long and winding. So far, very few regulatory-approved biomarkers have entered clinics [14,15,16].

The introduction of biomarker measurement into clinical practice is challenging. Several scientific, economic, and regulatory barriers need to be overcome before biomarkers can reach clinical practice [14,17,18,19]. Moreover, targeted adopters will respond heterogeneously to a new test [20]. As of yet, research focusing on identifying challenges at the implementation stage of the biomarker development pipeline has been scarce [21,22].

Different theoretical models have been developed to better understand users’ intention to accept a new technology [23]. One of these frameworks is the Non-adoption, Abandonment, and Challenges to the Scale-UP, Spread, and Sustainability of Health and Care Technologies (NASSS) framework, developed to theorize and evaluate challenges to the scale-up of health and care technologies [24,25]. This NASSS framework aims to detect the determinants of the implementation processes of complex technologies in healthcare in seven different domains. So far, it has been helpful in a range of applications [26,27,28].

In this study, we used the NASSS framework to investigate clinician-perceived barriers and facilitators to the adoption of NAFLD NITs. The tests were selected from a list of NITs that are evaluated in the LITMUS (Liver Investigation: Testing Marker Utility in Steatohepatitis) study, a multicenter project that aims to develop, validate, and qualify a defined set of biomarkers that can enable the detection of high-risk NAFLD patients [29].

## 2. Materials and Methods

### 2.1. Study Design and Registration

This exploratory study employed a cross-sectional design. Using a mixed-methods approach, we combined qualitative and quantitative methods to create a comprehensive picture of the adoption of influencing factors. The study protocol was made available through the Open Science Framework (https://osf.io/vhzkm, accessed on 8 May 2022) and is reported using the SRQR checklist (Appendix A) [30].

### 2.2. Theoretical Framework

We selected the NASSS framework to evaluate the adoption of the selected NITs because of its solid theoretical foundation and its focus on detecting determinants of the implementation processes of complex technologies in healthcare [24,25]. The NASSS framework lists potential determinants in seven domains: (1) the condition, (2) the technology, (3) the value proposition, (4) the adopters, (5) the organization, (6) the wider system, and (7) embedding and adaptation over time (see Figure 1).

### 2.3. Sampling and Consent to Participate

This study is focused on clinicians, as these important stakeholders in the healthcare system play a significant role in the implementation and dissemination of medical tests. Clinicians routinely request tests in clinical practice, interpret the results, and make clinical decisions based on them. We selected clinicians from multiple European countries, different specialties, and variable levels of experience in working with NAFLD patients through both purposeful and snowball sampling.

Initially, an invitation was sent to (1) a list, provided by the LITMUS consortium, of clinicians experienced with NAFLD care from different countries in Europe, (2) the Dutch NAFLD clinicians working group, (3) the French NAFLD clinicians working group, and (4) clinicians from Belgium, Scotland, Germany and England, identified through the researchers’ networks. All clinicians who participated in this study provided explicit consent before contributing and all answers were processed anonymously.

### 2.4. Data Collection

#### 2.4.1. Scoping Phase

The study consisted of two phases: a scoping phase and a data collection phase (Figure 2). In the scoping phase, four interviews with NAFLD care experts were conducted, to create an initial list of tests and to discuss the most relevant concepts of the NASSS framework. Thereafter, a screening survey with the initial list of seven tests was sent to 129 clinicians, to gain knowledge about the current clinical use of each of the tests. The final list was assembled based on the answers to the screening surveys (Appendix A). The results from the screening survey informed the development of the final questionnaire and interview guide in the main data collection phase.

##### Selected Tests

The final tests were selected based on the clinicians’ responses to the screening survey. See Appendix A for the number of tests that the respondents use for their NAFLD patients in their current clinical practice. To capture both facilitating and hindering factors, we selected three tests with different dissemination levels: from a well-disseminated test, which was used by all the respondents (FibroScan), to a relatively new marker that is not yet well integrated into clinical care and was reported by only two clinicians (PRO-C3).

FibroScan: FibroScan vibration-controlled transient elastography (VCTE) is a widely available ultrasound-based fibrosis test, which measures liver stiffness by estimating the velocity of propagation of a shear wave through liver tissue [31,32];Enhanced Liver Fibrosis (ELF) test: ELF is a moderately available serum biomarker panel, which consists of three components: type III procollagen peptide (PIIINP), hyaluronic acid (HA), and tissue inhibitor of metalloproteinase-1 (TIMP1) [33,34];PRO-C3: This procollagen-based marker is a relatively new serum biomarker with limited availability outside clinical trials. The procollagen type III N-terminal peptide (P3NP) is a by-product of the cleavage of procollagen III to produce collagen III [35].

Results of FibroScan and ELF are indicative of the amount of liver fibrosis while those of PRO-C3 are also indicative of fibrogenesis, the process of active fibrosis synthesis. All three have been studied across multiple liver diseases, including NAFLD [36,37,38].

#### 2.4.2. Data Collection Phase

##### Qualitative Data Collection

Participants who responded to our screening survey were invited to participate in a semi-structured interview of approximately 40 min, conducted online, with a pre-defined topic guide (Appendix A). The topic guide was piloted on one respondent. Revisions were made to the final version after the piloting interview and the first couple of interviews.

##### Quantitative Data Collection

To supplement the findings from the semi-structured interviews we also disseminated a quantitative questionnaire, designed based on the questions from the topic guide to the respondents. We offered a five-point Likert scale for responses, ranging from “strongly disagree” to “strongly agree”. We also invited clinicians to rate their level of knowledge for each test and to indicate their opinion about the most important barrier and facilitator for adopting each test using two open-ended questions (Appendix A).

### 2.5. Data Analysis

#### 2.5.1. Qualitative Data Analysis

All interviews were transcribed ad verbatim using transcription software Otter.ai version 2.1.41.612 (Lost Altos, CA, USA). Transcripts were analysed with the qualitative data analysis software ATLAS.ti version 9.1.2 (Berlin, Germany). Two authors (RE and YV) independently coded the first two interviews and discussed the codes. After reaching a consensus on the coding strategy, coding was completed using a combination of inductive and deductive thematic analysis and the NASSS domains as an analytical framework. RE led the analysis, and YV verified all codes, to achieve content conformity of the categories and themes. The quotations reported in the text were slightly edited to improve the readability while the meaning of the original texts was retained [39].

#### 2.5.2. Quantitative Data Analysis

Collected data were evaluated using descriptive statistics in R software version 3.5.2 (Vienna, Austria). Responses to each question are reported as: Disagree (scores 1 and 2), Neutral (3), and Agree (4 and 5).

## 3. Results

Thirty-nine of 129 invited clinicians responded to our screening survey (Appendix A). Twenty-seven filled in our questionnaire (Table 1); 16 of those also participated in the interviews (Figure 3). Participants were from eight different countries and spanned a wide range of experience (3–36 years), and specialties (Table 1). All clinicians were using FibroScan in their clinical practice, three also used ELF, while none had clinical experience working with PRO-C3.

Clinicians reported several factors that could influence the adoption of these three tests. These factors were categorized based on the respective NASSS domains. The complete list of identified barriers and facilitators is reported in Table 2, while the main factors are captured below.

### 3.1. Identified Factors Affecting Tests’ Adoption

#### 3.1.1. The Condition

Clinicians defined NAFLD as a multi-system disease highly associated with obesity, metabolic syndrome, or type 2 diabetes mellitus. They described that its complex pathogenesis and natural course are not fully understood and considered it essential to investigate disease progression in patients with different degrees of NAFLD activity.

According to most clinicians, the complexity of NAFLD would not hamper the adoption of the three selected tests (see Table 3). However, they believed that accurate diagnosis and management of such a complex disease would be challenging if using a single test.
“In our more complex diseases, where the decision and treatment depends a little bit on the assessment of [the] fibrosis, FibroScan alone will not be sufficient and have to be complemented with something else.”(Hepatologist)

#### 3.1.2. The Technology

Clinicians generally perceived the selected tests as easy to use in clinical practice. However, they mostly agreed that the test results are not always sufficient for decision making (Table 3). They specified that, practically, it is unlikely that a single test would be able to accurately rule in or rule out the disease. For this reason, they almost always use a combination of tests.

Concerns were also raised regarding the utility of the test. Almost all clinicians referred to the existence of a large grey zone, which impacts the discriminatory performance of these tests: they would only trust low and high values.
“So, there is this greyish area where the accuracy is not good enough. And the real low values give you a fair accuracy, there’s absence of significant fibrosis and certainly cirrhosis, and this higher limit where you’re sure that there is significant fibrosis and possibly even close to cirrhosis. And then there’s this greyish area where you’re not certain.”(Gastroenterologist)

The three selected tests were perceived to need different levels of knowledge for accurate performance in clinical practice. For FibroScan, for example, adequate training is required for proper implementation, while automated measurement of blood samples in laboratories can play a role as a facilitator for the adoption of blood-based markers. The human element in measurement with ultrasound-based tests, such as FibroScan, and the resulting intra-operator variability were believed to be factors that could challenge the current clinical practice and the test’s further adoption.

Furthermore, the results generated by the tests are not always easily interpretable. The complicated interpretation was mentioned as another critical factor influencing the usage of the tests, while the existence of a clear cut-off was believed to facilitate the interpretation and, consequently, test adoption.
“More important is how you interpret the results. And interpreting the result is a bit more complicated, and it requires knowledge of the tool and its pitfalls, particularly where you are likely to get false-positive results, and what to do when you suspect and therefore what to do as a result.”(Hepatologist)

Clinicians indicated an essential need for robust evidence of the adequate performance of the tests across different clinical settings. They stated that FibroScan and ELF have been substantially validated, while the available research is too limited to convincingly demonstrate PRO-C3 accuracy in detecting NAFLD progression. As these tests are used in different levels of care and clinical settings, which differ in the prevalence of at-risk NAFLD patients, a lack of studies that carefully consider these differences when estimating the performance of the test can considerably hamper adoption.
“At this stage, I don’t know how FIB-4, PRO-C3, and ELF perform in relation to each other and in different patient groups. I can imagine that advanced stages of NASH in certain patients such as morbidly obese patients or patients with diabetes will be better identified with the ELF test than with a FIB-4 test. This is something that we should investigate. We have very good data from the UK and other countries. But for example, we have no data in the Netherlands, our population could be different than the UK population. So we need to do more research and to establish the sensitivity and specificity in our population.”(Internist)


#### 3.1.3. The Value Proposition

Apart from substantial concerns regarding the benefits of new tests for specific populations or health settings, clinicians also referred to the high costs of the tests as one of the main barriers to the adoption of new NITs (Table 2).

Although high-quality cost-effectiveness analyses of these three tests are scarce and approved therapies for treating NAFLD are not yet available, all clinicians agreed that a NIT could be beneficial, for both clinicians and patients, by reducing the number of invasive and costly biopsy procedures. They also stated that all new NITs should be compared with less costly, already available tests, such as FIB-4, which has shown acceptable performance for ruling out advanced fibrosis (Table 4).
“So they could be cost-effective, but I don’t think they will beat using only FIB-4. The problem with all these cost-effectiveness analyses is that they are heavily influenced by what you put into them. And it depends on who you talk to how you really should count these costs. So even though cost-effectiveness analysis are very, very important, they are sometimes skewed, depending on which researcher that does them. But compared to the more normal or ordinary tests, both ELF and PRO-C3 are quite expensive. So that’s a barrier.”(Hepatologist)


#### 3.1.4. The Adopters

The participating clinicians reported different levels of knowledge about the three selected NITs (Appendix A). They stated that, while FibroScan is well implemented and widely used in most European countries, there is more limited understanding of the use of ELF and PRO-C3 in the NAFLD care pathway, which can significantly hinder the adoption of these tests (Table 2 and Table 4). The higher number of “neutral” answers for PRO-C3, compared to the other two tests, reflects the limited knowledge about this test that prevented the respondents from making strong statements about the test.

Differences in the level of knowledge among our respondents were not only observed between countries. Clinicians referred to this difference as a potential hampering factor that can affect test adoption across regions within the same country. They observed a considerable range in awareness in countries and settings where NAFLD is more prevalent. For instance, adopting a new NIT might be simpler in a tertiary care setting or an academic hepatology clinic in the UK than in a primary care setting in the Netherlands, with lower NAFLD prevalence.

The involvement of multiple stakeholders, from laboratory technicians to practitioners and hospital managers, makes implementation particularly challenging. Here, clinicians highlighted the important role of the so-called “local champions”. They stated that an enthusiastic clinician, who would initiate and closely monitor the adoption process, or a lab professional, willing to handle all practicalities in the lab, could significantly facilitate the adoption process of NITs.
“[as a clinician] if you come up with evidence, and as a group, or maybe even two clinics, in our case, vascular medicine, and hepatology, if you say this is important, then we’re going to have to be there in order to follow this development, or maybe even be leading in the development in the Netherlands, in order to convince boards of directors and insurance companies to provide financial support.”(Endocrinologist)
“[as a clinician] if you just come up with evidence, and as a group… in our case, vascular medicine, and hepatology, if you say this is important, then we’re going to have to be there in order to follow this development, or maybe even be leading in the development….[as] you’d have to convince [the] boards of directors to provide this funding.”(Endocrinologist)


#### 3.1.5. The Organization

Clinicians indicated that academic hospitals typically have more resources available to start using new tests. In almost all clinical settings, the initiative for adopting a new NIT would start with a clinician who can convincingly demonstrate the clinical need (local champion). Engagement with the management team and allocation of budgets were described as other important factors for successful implementation (Table 2 and Table 5).
“It was initiated by clinicians, who really wanted something to identify patients with these disorders, NAFLD and NASH, and we realized that the echography [Conventional ultrasound] was not sufficient, and that the usual algorithms are not sensitive enough. So we wanted a more precise measurement [such as] the FibroScan to take better care of our patients. So it was initiated by clinicians. And then we started talking to management. Together, we organized the money to buy the machine and it was very easily implemented in our center…”(Internist)


Most clinicians believed that extensive work is needed to properly adopt ELF and PRO-C3 in clinical practice, with less effort required for FibroScan, which is already implemented in many countries (Table 5). Changes might be needed in both intra- and extra-organizational routines, such as training sessions and changes in laboratory routines. Clinicians involved in implementing clinical care pathways with ELF and FibroScan mentioned other necessary changes at the extra-organizational level. They explained the need for a referral management system that would enable general practitioners to simply order a new test and guide them to interpret results properly, referring patients to secondary care when needed.
“That will be mainly at the level of the clinical lab where all the technical things needs to happen. So it should be integrated in the machinery of the lab and in the protocols of the lab. And there should be knowledge of the technical staff at the lab, if there is any specific manipulation needed, which is not done automatically by the machine. So it’s mainly a question of looking into the technical aspects of the lab and the training of staff at the clinical lab.”(Hepatologist)
“And what we have put out there in primary care is some very clear [guidance], there’s a big box that says ELF [score] below this; fine reassure [the patient], ELF at this level; fine refer [the patient onwards]. So it’s quite prescriptive in the sense, they don’t have to think too much about what the ELF components are and what it’s telling them. They’re just being guided by the result.”(Endocrinologist)


#### 3.1.6. The Wider System

Adoption of new NITs was perceived as a complex process, influenced by various national and international bodies. National expert groups, which include different specialists who actively collaborate to improve the care path for NAFLD patients, were named as one of the most important facilitators (Table 2). In addition, international consortiums and groups, such as the European Association for the Study of the Liver (EASL) were considered influential. Most of the clinicians agreed that current practice guidelines on NITs in NAFLD are sub-par (Table 5). FibroScan has appeared in many guidelines, but ELF and PRO-C3 are mostly absent.
“So cost-effectiveness are performed by people who aren’t even in that specialty area. So it’s a group of people who just look at the data: statisticians. And then there might be one advisor on the group who is from that area. It’s hard for them to argue sometimes against all these numbers.”(Hepatologist)


NAFLD care pathways were seen to differ widely across countries and regions. Referral and reimbursement systems, for instance, differ across health care systems. A lack of reimbursement was mentioned by most clinicians as the most important barrier hampering adoption (Table 2 and Table 5).

#### 3.1.7. The Future Outlook

Whilst most clinicians were optimistic about the future adoption of the three selected NITs on a larger scale, there were differing views on where to implement them (Table 5). The blood-based tests were perceived appropriate for primary care level screening. Some clinicians thought that the future of non-invasive testing for NAFLD lies in shear-wave elastography incorporated in ultrasound machines since every hospital has ultrasound machines already. Others believe that fibrosis assessment in NAFLD needs to be a dedicated procedure, not an ancillary measurement, that would best benefit in its overall interpretation from the knowledge and experience of a hepatologist.
“Ultrasound elastography is an elastography technique incorporated into regular ultrasound machines, which are available in all hospitals. I think this technique has the best chance to become the first line [test for] measuring fibrosis in NAFLD.” (Gastroenterologist)


## 4. Discussion

This explorative mixed-methods study highlights the complexities of NAFLD NITs’ adoption process and may contribute to a deeper understanding of the challenges and influencing factors involved. Different barriers and facilitators from the clinicians’ perspectives were reported for the adoption of FibroScan, ELF, and PRO-C3 in Europe (Table 2). Insufficient knowledge and awareness of NAFLD and NITs, a lack of practical guidelines built upon robust evidence for specific patient populations and care settings, and the absence of reimbursement were perceived as some of the most important barriers. Other factors, such as the presence of local champions, a proper functional payment system, and resources in academic hospitals were seen to play a facilitating role in the process.

### 4.1. Strengths and Limitations

The use of a structured and validated framework and integration of qualitative and quantitative methods contributed to a systematic data collection and synthesis and increased the credibility and reliability of the findings [40,41]. Semi-structured nature of the interviews allowed participants the freedom to express their views on their own terms. The questionnaire and the semi-structured interviews constituted data triangulation and consequently increased the study’s internal validity and confidence in the findings [42].

Nevertheless, it is important to consider the potential limitations of this study. Despite a large number of invited clinicians, a small number of clinicians participated in the interviews. In addition, our group of participants was selected using a purposeful sampling strategy and mostly from western Europe, which can limit the generalizability of the findings [43]. This study was not designed to represent all clinicians who work with NAFLD patients. We included clinicians from different specialties and various countries with varying levels of experience to improve the generalizability. However, the list of barriers and facilitators might differ in other countries, which were not reported in this study, due to different reasons such as differences in their health care systems. This study was focused on clinicians, as one of the most important stakeholders in the adoption process of NITs, to gain their in-depth perspectives. Future studies should be performed on the larger and more heterogeneous samples to get other stakeholders’ insights and experiences.

### 4.2. Implications for Practice and Research

Over the past decades, health care systems have experienced waves of rapid development, with new insurance models, regulatory changes, and novel technologies. In these evolving health systems, the development and implementation of NITs as new technologies for detecting specific health conditions received a great deal of attention. A number of other studies have evaluated the introduction and dissemination of NITs in different medical fields to enhance understanding of the diseases and extend the monitoring and treatment options [28,44,45,46]. Some also evaluated various factors that can influence the adoption of these tests [46].

Multiple NITs have been developed and evaluated in the hepatology field. In parallel with a steady increase in the incidence of NAFLD at the global level, interest in these NAFLD NITs has grown. Several regional guidelines were published for the clinical management of NAFLD, but many health care settings still suffer from a lack of written practical pathways to identify patients and link them to care pathways [47,48], which makes diagnosing NAFLD an enduring challenge.

There have been many discussions about the preferred NAFLD care pathway, without a clear consensus on the NITs to be used [47]. This debate could influence practitioners’ decisions about including new tests in their routine clinical practice, as well as decisions at the hospital management level about adoption.

Our findings show that clinicians’ acceptance and adoption of new NAFLD NITs highly depend on a test’s ability to improve patient care in clinical practice. Robust evidence is required, demonstrating that the performance of new tests is substantively superior to that of already existing approaches, while the strong influence of differences in the pre-test probability and comorbidities on the performance of the test is sufficiently considered [36,47,49].

There is a lack of adequate head-to-head comparisons of new tests in clinical settings that differ in disease prevalence and patient characteristics, including primary care settings. This may contribute to the limited awareness and knowledge about these tests, consequently affecting their usage. In addition, this absence may also hamper the development of practical guidelines that define optimal disease diagnosis, patient management strategies, and a successful regulatory approval process.

Besides the need for robust evidence and clinical guidance, the high costs associated with many new diagnostic tests in different medical fields also create significant barriers to their widespread adoption [50,51]. The costs of the NAFLD NITs and their coverage by health insurance vary considerably between countries. When a new test is not advocated sufficiently by clinical guidelines, it is more likely not to be reimbursed. Reimbursement will often depend on convincing evidence in guidelines. This evidence-based healthcare paradigm [52,53] also affects local budgeting for NALFD NITs. Academic hospitals may be at an advantage, due to their ability to attract research grants, which bring more opportunities for adopting new tests for research purposes.

As in any other field of technology, the crucial role of local champions in the adoption of NAFLD tests should not be underestimated [54]. These clinicians with a special interest in and knowledge about a new test are vital for initiating the adoption process, increasing awareness and improving knowledge of NAFLD and NITs. As such, they can significantly facilitate the adoption process and influence functional reimbursement systems that serve the interests of both the health care system and patients.

## 5. Conclusions

Clinicians consider the adoption of new diagnostic NAFLD tests a complex process, one that can be promoted or restricted by several factors, such as robust evidence, practical guidelines, an adequate payment system, and local champions. Identifying these influencing factors helps clinicians and health decision-makers to identify areas for improvement in the test’s adoption process and more effectively implement new tests in clinical settings.

## Figures and Tables

**Figure 1 jcm-11-02707-f001:**
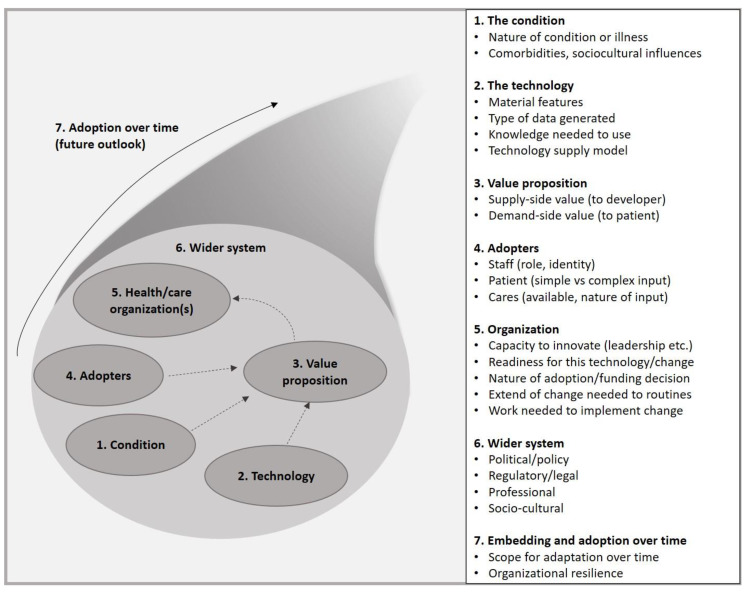
NASSS (Non-adoption, Abandonments, Scale-Up, Spread, and Sustainability) framework.

**Figure 2 jcm-11-02707-f002:**
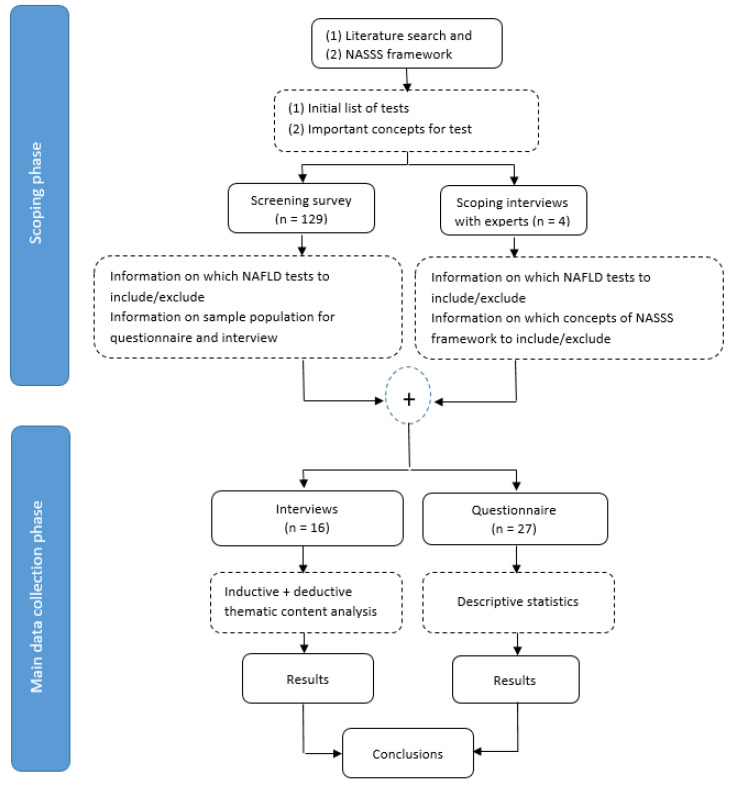
Study design flow diagram.

**Figure 3 jcm-11-02707-f003:**
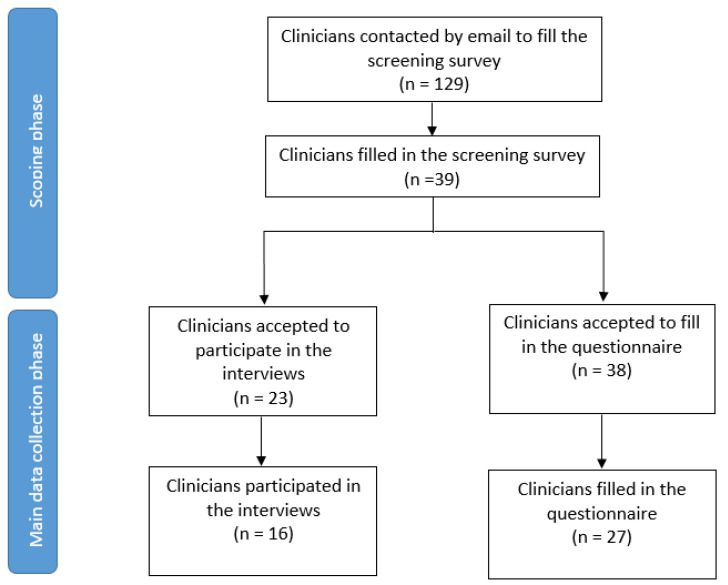
Flow diagram of the respondents.

**Table 1 jcm-11-02707-t001:** Baseline characteristics of questionnaire respondents and interview respondents.

	Questionnaire Respondents (N = 27)	Interview Respondents (N = 16)
**Age (Years)**		
Mean (Range)	43 (30–68)	46 (30–68)
**Country of practice**		
Belgium	6	3
UK	6	4
France	4	3
Germany	3	0
Greece	1	0
Italy	2	1
Netherlands	5	4
Sweden	0	1
Specialty		
Endocrinology	2	2
Gastroenterology	5	2
Hepatology	15	8
Internal medicine	5	4
**Years of experience (Years)**		
Mean (Range)	16 (3–36)	19 (3–36)

**Table 2 jcm-11-02707-t002:** Summary of the main facilitators and barriers sorted under the Non-adoption, Abandonment, Scale-up, Spread, and Sustainability (NASSS) framework’s domains.

Domain	Barriers	Facilitators
**1.The condition**	Multisystem disease linked with other extra-hepatic chronic diseases	
**2.The technology**	Difficult interpretation	Robust clinical evidence
	Long-time interval between measurement and access to the test’s result	Proved better performance in detecting the target condition compared to other available tests
	Need for extra training	Quick measurement process and data generation process
	Availability in different clinical settings	No need for specialist to perform the test
	Usage for research purposes	Easy access to the test in the clinical setting
	Lack of empirical evidence	Availability of the test for the research purposes in an academic clinical setting
	Low performance as a single biomarker-based test	Knowledge needed for proper interpretation of the test results
	Inter operator variability	Possibility of using test for other target conditions in clinical pathway
**3.The value proposition**	Non-existence of a therapeutic intervention	Comprehensible results for patients
	Higher costs compared to existing tests	Lower costs compared to existing tests
	Doubting quality and appropriateness of the test for specific population or health setting	No need for extra sampling- possibility of measuring the biomarker using the samples collected for routine measurements
**4.The adopters**	Involvement of multiple adopters in the implementation process	Local champions-interested clinicians or lab professional
	Inconsistent Acceptance	Small workflow changes-simple ordering method for the clinicians
	Acceptance of a new test and changing the routine clinical approach by clinicians	
	No sufficient awareness about non-alcoholic fatty liver disease (NAFLD) and non-invasive tests	
**5.The organization**	Available funding	Sufficient intra-organizational financial support
		Support from management team
		Already implemented similar devices
**6.The wider system**	Lack of reimbursement	Proper reimbursement system
	Health system local differences	Local and national disease specialist group and scientific consortiums
	Absence of practical national guidelines	
**7.The future outlook**	Complicated, costly and time consuming process for future implementations	

**Table 3 jcm-11-02707-t003:** Results per items of “The Condition” and “The Technology” domains of the Non-adoption, Abandonment, Scale-up, Spread, and Sustainability (NASSS) framework as reported by responders.

	ELF (N = 27)	FibroScan (N = 27)	PRO-C3 (N = 27)
**The Condition**			
**The complexity of NAFLD as a disease hinders the adoption of the test in clinical setting**			
**Agree**	7	6	8
**Neutral**	9	4	10
**Disagree**	11	17	9
**If NAFLD cases weren’t as diverse as they are, adoption of the test in clinical setting would be easier**			
**Agree**	8	7	8
**Neutral**	9	7	9
**Disagree**	10	13	10
**The Technology**			
**A lot of training and support is needed to use this test ***			
**Agree**	4	12	3
**Neutral**	8	5	9
**Disagree**	14	10	13
**The data generated by this test are not always used for decision making in NAFLD clinical practice ***			
**Agree**	8	6	11
**Neutral**	12	3	10
**Disagree**	6	18	4
**The data generated by this test are not always sufficient for decision making in NAFLD clinical practice ***			
**Agree**	15	15	13
**Neutral**	7	3	8
**Disagree**	4	9	4
**The data generated by this test are always trusted ***			
**Agree**	3	6	3
**Neutral**	7	5	7
**Disagree**	16	16	15
**Overall, this test is easy to use**			
**Agree**	13	21	9
**Neutral**	8	4	9
**Disagree**	6	2	9

* Including missing. NAFLD, Non-Alcoholic Fatty Liver Disease; ELF, Enhanced Liver Fibrosis; PRO-C3, procollagen type III.

**Table 4 jcm-11-02707-t004:** Results per items of “The Value Proposition” and “The Adopters” domains of the Non-adoption, Abandonment, Scale-up, Spread, and Sustainability (NASSS) framework as reported by responders.

	ELF (N = 27)	FibroScan (N = 27)	PRO-C3 (N = 27)
**The Value Proposition**			
**This test is not a cost-effective option for the organization I work at ***			
**Agree**	5	4	6
**Neutral**	13	2	16
**Disagree**	9	21	4
**From my perspective, this test is more advantageous regarding patient management over existing NAFLD clinical practice (i.e., liver biopsy) ***			
**Agree**	10	20	7
**Neutral**	11	5	12
**Disagree**	5	2	6
**This test has an added value for me as a clinician ***			
**Agree**	13	26	9
**Neutral**	11	1	14
**Disagree**	3	0	3
**This test has an added value for the patient ***			
**Agree**	12	25	10
**Neutral**	13	1	14
**Disagree**	2	1	2
**The Adopters**			
**The use of this test changes the usual practice of my work as a clinician for NAFLD care in a positive way ***			
**Agree**	6	25	3
**Neutral**	15	1	16
**Disagree**	6	1	7
**Patients are not always willing to cooperate with this test ***			
**Agree**	5	3	4
**Neutral**	10	2	11
**Disagree**	12	22	10
**There is not enough understanding of the use of this test in the pathway of decision making for NAFLD care ***			
**Agree**	13	5	16
**Neutral**	6	6	7
**Disagree**	7	16	3

* Including missing. NAFLD, Non-Alcoholic Fatty Liver Disease; ELF, Enhanced Liver Fibrosis; PRO-C3, procollagen type III.

**Table 5 jcm-11-02707-t005:** Results per items of “The Organization”, “The Wider System”, and “The Future Outlook” domains of the Non-adoption, Abandonment, Scale-up, Spread, and Sustainability (NASSS) framework as reported by responders.

	ELF (N = 27)	FibroScan (N = 27)	PRO-C3 (N = 27)
**The Organization**			
**There is not enough support and advocacy in the organization for the adoption of this test ***			
**Agree**	12	3	13
**Neutral**	9	2	11
**Disagree**	5	21	2
**There are enough time and resources in the organization for the adoption of this test**			
**Agree**	12	18	8
**Neutral**	4	0	9
**Disagree**	11	9	10
**here is not enough allocated budget in the organization for adoption of this test**			
**Agree**	18	10	14
**Neutral**	8	5	12
**Disagree**	1	12	1
**There is a shared vision in the organization between management and clinicians regarding adoption of this test***			
**Agree**	3	12	2
**Neutral**	13	9	16
**Disagree**	11	6	8
**Extensive work is needed to properly adopt this test in clinical practice**			
**Agree**	14	9	17
**Neutral**	8	3	7
**Disagree**	5	15	3
**The Wider System**			
**The qualification requirements and regulatory landscape for this test are well-defined in my country ***			
**Agree**	2	15	0
**Neutral**	9	6	8
**Disagree**	16	6	18
**Currently, there are no rigorous clinical guidelines for use of this test in my country**			
**Agree**	16	7	20
**Neutral**	3	1	3
**Disagree**	8	19	4
**There is enough reimbursement available for this test in my country ***			
**Agree**	7	6	6
**Neutral**	9	6	10
**Disagree**	11	15	10
**The Future Outlook**			
**The test has the potential to be adopted at larger scale for NAFLD care in the future ***			
**Agree**	15	26	14
**Neutral**	7	1	10
**Disagree**	5	0	3

* Including missing. NAFLD, Non-Alcoholic Fatty Liver Disease; ELF, Enhanced Liver Fibrosis; PRO-C3, procollagen type III.

## Data Availability

The datasets used and analyzed during the current study are available from the corresponding author on reasonable request.

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
