# Peer review of "Clinicians’ Perspectives on Barriers and Facilitators for the Adoption of Non-Invasive Liver Tests for NAFLD: A Mixed-Method Study"

_jcm, 2022, doi:10.3390/jcm11102707_

Round 1

Reviewer 1 Report

This study investigated the clinicians’ perspectives on the barrier and facilitation of the adoption of non-invasive liver test methods for non-alcoholic fatty liver disease. The data were collected by the NASSS framework by analysis of seven factors. Overall, the study was well-performed and discussed. One major limitation is that the case size shown in Figure 3 is small. In addition, there are some minors. The words in Figure 1 are dim, recreate this figure. The style of references requires a revision according to journal type, and remove line 589, the last line of the manuscript.

Reviewer 2 Report

This is a very interesting and relevant study lacking in the literature. This way, I congratulate the authors for this initiative since NAFLD is a substantial public health problem that needs studies like the present one to improve the awareness of clinicians and stakeholders to improve its management with the best possible accuracy. Its main strength is that it reflects real-life barriers regarding the general use of NIT in NAFLD, which is of utmost importance. However, I would like to discuss some issues in order to improve the quality of the study and maybe draw the attention to the authors to validate it worldwide, helping to diminish the enormous gap that exists in NAFLD management in the different regions of the planet. ,
Methods: Although the authors have shown in the supplementary archives the selected NIT ranked among participants, I suggest these were described in the manuscript, as well as the ranking result among the different tests in the results session, making clearer why Fibroscan, ELF and Pro-C3 were included in the sequential interview. It was a surprise that FIB-4 was not ranked since it is an available and cheap test included in many guidelines to exclude advanced fibrosis in NAFLD patients. Also, it was a surprise to include Pro-C3, which is rarely used and available, as shown in the results. Even in the introduction session, Pro-C3 is not mentioned (only FibroScan and ELF). Please clarify.
The authors invited only clinicians from Western Europe, making the study rather frail. The selected countries were those with a higher income and developed from an economic standpoint. My suggestion is that if the authors want a precise picture of the limitations of using NIT in Europe, the study should be more inclusive. Please discuss this point. 
Results:
Only 30% of physicians accepted to answer the screening survey (39/129). According to table 1, there was a heterogeneous representation among the different regions in Europe regarding the participants. According to table 1, there were no clinicians from Spain, Portugal or Western Europe. Were these clinicians invited? If not, please justify. This could cause a selection bias and might be, as already said, another limitation to discuss. 
In most of the answers, there was a high proportion of "neutral" answers. How did the authors interpret this result? This could be discussed since the authors can verify if there was homogeneous neutrality regarding countries or clinicians' specialties (hepatologists? gastroenterologists? internists?). Could the authors revise this item and include this item as another limitation of the study?
